# Cultural and Religious Diversity in Early Childhood Education Implications of Socialization and Education for the Geographies of Childhood

**Christoph Knoblauch** 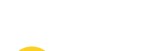

Institute for Theology, Ludwigsburg University of Education, 71634 Ludwigsburg, Germany;
christoph.knoblauch@ph-ludwigsburg.de

**Abstract:** Cultural and Religious Diversity in ECE is discussed from multiple perspectives and influenced by different parameters. In this context, culturally and religiously sensitive education faces various dynamic and conflictual challenges, such as different comprehensions of the concepts of culture and religion or current transformations in society. Social spaces such as kindergartens play a major role in offering the potential for diversity to be experienced and reflected in the context of socialization and education. Focusing on the manifold relations between cultural and religious diversity in education, this paper discusses evaluation findings from a qualitative study in the German Early Education sector. Perspectives from children, parents, and educators on the implications of socialization and education for the geographies of childhood are presented with a special focus on options for culturally and religiously sensitive education. The empirical findings, therefore, focus on experiences and assessments from different participants in the field of ECE, offering a multiperspective view on the topic. The study uses semi-structured qualitative interviews and content-based evaluation, interviewing over 200 children, educators, and parents in group interviews. The results are discussed in categories such as socialization of children, competencies of educators, underlying pedagogical concepts, experiential learning, and others. In this context, the paper offers an in depth discussion of the potential of a culturally and religiously sensitive education and the role of communities and religious institutions. Against this backdrop, the question discussed is how these results can be constructively implemented to improve the constructive perception and the usage of educational spaces in ECE.

**Keywords:** culturally and religiously sensitive education; early childhood education; socialization; multiperspective qualitative study; social and educational spaces

## 1. Introduction

Cultural diversity and religious diversity influence social coexistence in societies all over the world (Mellor 2004). People with different cultural and religious backgrounds and diverse histories of socialization live together in various contexts and share their individual religious and cultural concepts with their community. Interrelations of society, culture, and religion can be observed in these contexts in various ways, as individuals develop beliefs, worldviews, and traditions based on socialization, education, and experiences (Geertz 1973). Paul Tillich describes culture as one form of the expression of religion. At the same time, he regards religion as a content of culture (Tillich 1962). "Religion is part and parcel of the culture in which it is lived" (Witte 2001). These interrelations not only show up in families and societies but also in various aspects of everyday life and human relationships. In times of globalization and migration, the importance of cultural and religious identities, and their visibility within society, have to be appreciated in order to embrace diverse cultural and religious affiliations constructively (Beyers 2017). This produces a need for sensitive learning environments, which have to be built on principles

of appreciation and equality (Auernheimer 2012). Learning involves various processes that are almost always embedded in lived relationships and are therefore marked by individual views and concepts (Boschki and Biesinger 2008). Ambivalent concepts of culture and religion and the resulting worldviews and practices demand a high level of resilience, often described as tolerance of ambiguity (Frenkel-Brunswick 1948). Against this background, the sensitive inclusion of socialization contexts becomes a crucial responsibility of educational institutions (Gaus 2021). Early childhood education (ECE) plays an important role in the constructive inclusion of diversity, as children with diverse backgrounds come together outside their family contexts. A child's early years can become the foundation for future attitudes. ECE institutions are social and democratic spaces of learning and are therefore closely interlinked with cultural and religious contexts (Stockinger 2018). Kindergartens and other ECE institutions, therefore, have the responsibility to offer spaces for learning and growing up that are characterized by an appreciation for diversity and equality. Cultural and religious responsiveness and the willingness of the educator to establish an open and sensitive climate seem to be important traits for these learning spaces, which have to include the various geographies of the child's socialization (Sanders and Downer 2012).

This paper discusses the results from a multiperspective qualitative study with a special focus on the possible potential for culturally and religiously sensitive education in ECE (Knoblauch 2019). In this study, children, parents, and educators share their ideas and experiences regarding cultural and religious diversity in learning spaces and discuss the role of kindergartens. The paper displays (Section 2) the theoretical framework of the topic concerning a constructive approach to education in general and diversity education in particular. Having presented the theoretical approach, this paper then presents (Section 3) the design of the study and (Section 4) its core findings. The discussion of these findings (Section 4) reveals the importance of culturally and religiously sensitive education in ECE, reflects on implications for early childhood institutions and shows possible potential for diverse learning spaces.

## 2. Theoretical Aspects of Cultural and Religious Diversity in ECE

### 2.1. Interactive Constructivism in Education

The child's mindscape develops through intense and active processes of construction and interpretation in the interaction with others (Reich 2007). In diverse learning contexts, the child is confronted with various cultural and religious ideas and traditions that might be different from its individual socialization contexts (Willems 2008). Interaction with others challenges the learner to acknowledge and deal with diverse social backgrounds, and to appreciate various geographies of socialization (Grümme 2017). Within these processes, the child, actively and passively, finds itself in different roles: it acts as an (1) observer, a (2) participant, and an (3) agent in the construction of competencies and attitudes (Reich 2003). (1) The child observes others and itself as it meets other children and educators. Thus, it experiences similar and different cultural and religious beliefs and practices, and reflects on these. (2) Furthermore, the child acts as a participant and shares individual ideas and beliefs by interacting with others. As participants, children share their identities (Bauman 2000), expressing individual cultural and religious ideas according to their age and abilities. (3) As an agent, the child becomes an active subject that expresses ideas and follows individual goals. These ideas and goals can serve as the foundation for new interactions if being responded to by others. Previous observation and participation are the foundation of this expression. As an observer, participant, and agent, the child constructs, as an active learning individuum, new ideas and beliefs. Involvement in an inspiring learning environment and interaction with peers and educators can help children constructively develop their potential for self-education (Weltzien and Viernickel 2008). Thus, children need learning environments that enable them to observe other beliefs, interact with others who share different histories of socialization, and act as agents, expressing individual views. This approach of interactive constructivism requires a child-centered approach, which takes into account the highly individual and diverse geographies



of children's socialization (Doddington and Hilton 2010). Children have to be appreciated as active designers of their individual worldviews, which are developed through processes of interaction and in environments that foster relationships. Against this background, Humboldt's idea of alienation from one's own beliefs as a constructive means of learning plays an important role: in modern educational theory, the understanding of education often refers to Humboldt's fragment "Theorie der Bildung des Menschen" (Theory of the Education of Man) (Humboldt 1793). Learning potential evolves when children discover a world that is different from their life worlds. "Our encounters with the world are transformative in that they are mediated by 'self-alienation,' that is, alienation from our taken-for-granted and habitual self-understandings. [ . . . ] Bildung refers to the active and receptive self–other relation implicit in [the] educational process" (Knoblauch et al. 2021). Children have to be seen in the contexts of their sociality and relationships if education wants to be sensitive to cultural and religious diversity. In this regard, culturally and religiously sensitive education needs a theory of education that includes dimensions of constructivism, child-centeredness, and relationships.

### 2.2. Relationships as a Crucial Factor in Diverse Learning Environments

The discussed study focuses on dialogue and interaction between children in ECE contexts. Therefore, children's ideas and experiences play a crucial role and are discussed against the backdrop of learning arrangements and content in everyday situations in kindergartens (Herrmann 2019). The study is based on an educational approach that focuses on the lifeworld experiences of children and reflects these under the idea of relation-oriented learning (Everington et al. 2011). Therefore, personal, social, familial, institutional, temporal, and existential forms of relationships are taken into account. Subjectification always needs a relationship, which has to be based on a trustful encounter, and which is interconnected to various dimensions of life (Gößling 2010). Children's lifeworlds are therefore understood as worlds of relationships. Children's views and practices are understood in direct interconnection to their relationships (Boschki 2003). Against this backdrop, the term 'relationship' can be discussed in the context of different dimensions that are crucial for a culturally and religiously sensitive education:

(1) Relationship with oneself: From the perspective of social sciences, the term 'identity' in the first place refers to the child's relationship to itself, and this relationship is in a state of constant transition and inherits the competency to reflect on one's individual ideas and beliefs.

(2) Relationship with others: Identity is formed through inner and outer influences. The relationship with oneself develops in relationships with others, and interaction is crucial to this dimension as it leads to the development of the relationship with others—kindergartens, for example, can offer a foundation for this development, as children from diverse backgrounds interact.

(3) Relationship with social environments: Relationships in the social environment play an important role when encounter becomes a reciprocal exchange of ideas and experiences, and this is especially important when children learn in culturally and religiously diverse environments.

(4) Relationship with time: As cultural and religious traditions and practices are often connected to certain times and as existential questions are part of one's lifeworld, this dimension is crucial to relationships.

(5) Relationship to a higher being: This dimension discusses an existential relationship inherent to everybody and not necessarily connected to a religion, and children can experience this relationship as a connectedness to the world and can observe diverse interpretations of this relationship in their learning environment.

Thus, the theoretical framework of the discussed study builds upon ideas of a child-centered and relation-based education: "Children bring traditional practices, values, beliefs and the experiences of family and community to early childhood programs. Their sense

of inclusion increases in environments that allow their full participation and promote attitudes, beliefs, and values of equity and democracy" (McCain et al. 2011).

*2.3. Cultural and Religious Diversity in ECE: Challenges and Potentials*

Article 14 of the 'UN Convention on the Rights of the Child' (United Nations 1989) establishes the right of the child to freedom of thought: "States Parties shall respect the right of the child to freedom of thought, conscience, and religion" (Convention on the Rights of the Child, article 14). Article 29 emphasizes that education should promote: "(c) The development of respect for the child's parents, his or her own cultural identity, language and values ( . . . ) (d) The preparation of the child for responsible life in a free society, in the spirit of understanding, peace, tolerance, equality of sexes, and friendship among all peoples, ethnic, national and religious groups and persons of indigenous origin" (Convention on the Rights of the Child, article 29). These rights can be interpreted as a foundation for the sensitive appreciation of cultural and religious diversity in ECE, as they emphasize the importance of unbiased education. "Awareness of difference in religious education is a prerequisite for educational work that promotes the sensitivity and ability of children, young people, and adults to learn about differences" (Stockinger 2018).

Against this backdrop, concepts of anti-bias education (Derman-Sparks 2010), diversity education (Chan 2017), and culturally and religiously sensitive education (Weber 2014) have been analyzed to establish a framework for the evaluation of the data. The following principles can be seen as a first common ground for a culturally and religiously sensitive education in ECE, even though further categories will be needed:

(1)   The lifeworld of the child and the various dimensions of socialization have to be the starting point of culturally and religiously sensitive education.
(2)   Every ECE institution has to focus on diversity on a conceptual level, to guarantee a pedagogical foundation for children, educators, and parents.
(3)   ECE institutions are spaces where children can actively explore and experience cultural and religious diversity.
(4)   Every form of discrimination has to be identified and children have to be empowered to act against discrimination.
(5)   The child's lifeworld has to become a part of the learning environment, as every child should be encouraged to bring individual cultural and religious ideas and practices to the institution.
(6)   Children are enabled to experience cultural and religious diversity through encounters with other children, educators, parents, and partners in the community (such as clubs, cultural and religious communities, nursing homes, or schools).
(7)   Educators reflect their individual cultural and religious views and practices in the context of diversity.

The above principles are used to structure the data, as they lay a foundation for the development of the deductive categories.

## 3. Multiperspective Qualitative Research on Cultural and Religious Diversity in ECEC: Background of the Study

The study is part of a 4-year long research project on cultural and religious diversity in the ECE sector in Germany and was conducted at Freiburg University of Education, Tübingen University, and Ludwigsburg University of Education. The study mainly focuses on the potential for the development of cultural and religious competency and diversity competency in children between two and seven years. Therefore, the study discusses essential theories from developmental psychology, education, sociology, and theology in interdisciplinary discourse. Broad objectives include: (1) the discussion of interdisciplinary perspectives; (2) empirical research on potentials for competency development; and (3) the development of recommendations and a concept that can be used for (a) further discussion in the scientific community, and (b) for educators in the field. This paper focuses on the

empirical research, with a special focus on the implications of socialization and education for the geographies of childhood and diversity.

### 3.1. Design of the Empirical Study

The study uses methods of qualitative interview research (Berg and Lune 2017) and content analysis (Bengtsson 2016) in various ways and with a multiperspective design (Larkin et al. 2019). All interviews were transcribed, structured, and interpreted by the research team (Knoblauch 2019). The complex research focus—potentials for the development of cultural and religious competency and diversity competency—asks for an innovative approach, observing multiple perspectives through dialogue-based interviews including all partners in the education process: children, educators, and parents. The multiperspective approach takes into account that educational processes in ECE have to be discussed with perspectives from educators and parents as well, given the fact that they are crucial to the child's socialization (Bronfenbrenner 2009). Additionally, the perspectives of educators and parents can add new impulses to the data and are used to discuss the children's answers through means of triangulation (Vogl et al. 2019). In this context, over 200 children have been interviewed in small groups (2–4 children) and over 40 educators and parents have been interviewed in individual sessions. The selection of the sample was random and reflects the heterogeneous situation in German institutions. The interviews were semi-structured qualitative interviews using additional methods such as short stories, pictures, and items to encourage communication. The questions focused on experiences with and concepts of cultural and religious diversity. Festivals, places, communities, value concepts, existential questions, and other themes were discussed. The samples were framed with the method of 'purposeful sampling' (Palinkas et al. 2015), and they form a selection of information-rich cases related to the phenomenon of cultural and religious diversity. Therefore, samples from ECE institutions from cities (100,000–3,500,000 inhabitants) were used and the group interviews usually consisted of children with diverse cultural and religious backgrounds. The semi-structured interview situations offered the possibility to discuss questions in deep, clarify ambiguities, and develop a constructive dialogue between the interviewer and the participants. The multiperspective qualitative approach offers insights into (a) possible potential for the development of cultural and religious competence as well as diversity competence in children, (b) processes of socialization, (c) perspectives on cultural and religious diversity in ECE, (d) challenges and potential of cultural and religiously sensitive education in ECE, and (e) the possibility to review results from different perspectives. In these processes, the study follows established structures of qualitative research and analysis in manifold ways.

### 3.2. Findings of the Study

The qualitative study focuses mainly on the experiences and attitudes of children, educators, and parents. The analysis and discussion of the data show different combined categories, which are based on the deductive categories and new inductive impulses found within the data. The findings are structured and discussed according to these categories. Several answers and reflections show links to more than one category and are therefore discussed in various contexts.

**(a) The lifeworld of children and dimensions of socialization in the light of culturally and religiously sensitive education**

Regarding the results, the study indicates that children bring various individual experiences with cultural and religious contents or connections into ECE institutions. The interviewed children discuss a broad variety of (1) topics and (2) experiences in the context of cultural and religious themes. (1) Topics such as 'justice', 'equity', 'trust', 'belief', 'values', 'fear', 'hope', 'joy', and 'loneliness' are discussed by children, and are often connected with cultural and religious themes. Religious stories seem to be especially interesting to many of the interviewed children, who report their reflections on the above topics on the grounds of stories from religious scriptures. (2) The discussed experiences are often based

on certain events within the family, the community, or religious group. Some children discuss cultural and religious festivals as topics that they bring into ECE institutions, and which they also want to celebrate there. The results from educators and parents are similar, and underline the importance of a child-centered approach that takes into account the lifeworld of the children.

Regarding perspectives on socialization, the study indicates that families, social environments, and religious communities have a strong impact on children's views and experiences, and cultural and religious practices that are exercised in the family are, especially, taken into ECE institutions by children.

Regarding discussion of an example, the religious festival of Christmas is discussed by children, parents, and educators, as it is a popular Christian festival that is celebrated in many families and is visible in German society. Children report about family traditions and practices that are partly cultural and partly religious. Almost all children know about the festival and report on various experiences. Parents report family traditions connected to Christmas, and many educators celebrate Christmas in ECE institutions together with the children. However, the degree to which Christmas is celebrated as a religious festival differs: some families visit a church and reflect on biblical stories, while others celebrate Christmas as a cultural holiday without religious connotation.

From this discrepancy, the question arises about which role religious festivals should play in ECE institutions. On the one hand, children's experiences have to be included in ECE, but, on the other hand, children must not be forced to celebrate religious festivals. One solution might be to offer important religious festivals in ECE institutions with different levels of commitment. Children should be able to choose if they actually want to discuss affiliated topics of a religious festival and maybe take part in according practices, or join a different group that discusses the festival from a purely cultural perspective. Children should be offered various degrees of involvement to guarantee that they can choose levels of commitment. Thus, diverse lifeworlds can be included in ECE institutions without excluding others.

Regarding outlook, a culturally and religiously sensitive approach needs to include the life worlds and experiences of all children. ECE institutions should therefore be open to various cultural and religious backgrounds and include them in the learning environment. However, children must not be forced to participate in learning arrangements and have to be offered various degrees of commitment. Children need different opportunities: for example, (1) to learn about cultural and religious festivals, (2) to actually experience them, or (3) even help to celebrate them.

**(b) ECE institutions as spaces of cultural and religious diversity**

Regarding results, the study indicates that ECE institutions are naturally spaces of cultural and religious diversity due to the diverse backgrounds of children and educators. Additionally, various cultural and religious institutions show interest in collaborations with ECE institutions and hence become partners in the educational processes. Children, educators, and parents report on manifold topics and events in their institutions, where cultural and religious diversity can be seen and experienced. Children vividly discuss stories and festivals from different cultures and religions in their institutions. Furthermore, educators report that curricula for ECE discusses cultural and religious diversity as topics for ECE institutions. In these reports, concepts of culture and religion are often blended.

Regarding perspectives on socialization, cultural and religious communities seem to play an important role in this context as many children and educators report on festivals that are celebrated in collaboration with communities. Out of these collaborations, manifold learning options are developed.

Regarding the discussion of an example, when the interviewed children report about the 'Harvest Festival, they discuss cultural and religious aspects at the same time. While some children report a festival at a church or religious stories in their kindergartens, others discuss agricultural aspects and the change of seasons. Religious communities and non-religious clubs offer possibilities to celebrate this day and sometimes collaborate with ECE

institutions. Children and parents with different cultural and religious backgrounds report on this festival and emphasize different priorities for their celebration. Educators report on different collaborations, which focus on quite religious or rather cultural traits.

Regarding the outlook, ECE institutions reflect the cultural and religious diversity of the communities they are situated in. As the first organizations that children visit regularly, ECE institutions can have a strong impact on the child's socialization. In kindergartens, children are confronted with cultural and religious diversity and experience differences in various forms of interaction. ECE institutions are spaces of experienced diversity. Therefore, they have to develop diversity-sensitive approaches that offer children constructive and flexible learning options. The experience of diversity should also go hand-in-hand with possible experiences of belonging. Hence, children should find opportunities for identification. In view of ECE institutions as spaces of diversity, a culturally and religiously sensitive education should not be postponed.

**(c) Culturally and religiously sensitive education on a conceptual level**

Regarding the results, this study and the subsequent study from Weber (2014) indicate that (1) culturally and religiously sensitive approaches are rarely discussed explicitly in ECE institutions but, at the same time, (2) concepts of ECE institutions are often revised due to diverse learning groups and their potentials. (1) Most interviewed educators report that culturally and religiously sensitive approaches are not explicitly discussed in their ECE institutions. (2) However, concepts of ECE institutions seem to be undergoing constant change and have individual histories of development. Most educators report on a certain pedagogical concept, which serves as a foundation for the individual concept of their ECE constitution. The individual concepts are strongly influenced by the diversity dimensions experienced in the individual institutions. Certain topics seem to be of great importance to most interviewed educators when the concepts of their institutions are discussed in the context of diversity: collaboration with parents, the inclusion of the children's lifeworld, focus on the children's interests, a variety of learning options for the children, and experiential learning options.

Regarding perspectives on socialization, the interviewed educators mostly mention parents as a crucial factor in the cultural and religious socialization of children. The ECE institution is discussed as an important factor for socialization next to the parents.

Regarding discussion of an example, as the collaboration with parents seems to be an important factor in the concepts of ECE institutions, the individual cultural and religious backgrounds of the parents need to be included. One educator reported that a Jewish parent came to the institution several times and discussed the concept of the 'Sabbath'. This was experienced as a very authentic approach, as individual traditions and practices were explained by the parent. The children were able to learn about the concept of the 'Sabbath' and, at the same time, experienced an authentic insight from a person of reference. The educator reports that it was very important to discuss topics and methods with the parent beforehand to guarantee apt learning options.

Regarding outlook, culturally and religiously sensitive approaches have not yet been fully included in concepts of German ECE institutions. However, many concepts show culturally and religiously sensitive approaches that are developed by educators due to diverse learning groups in their institutions. It shows that concepts are undergoing constant change, which is driven by the needs of the learning groups. Educators seem to appreciate the possibility of developing the institution's concept. As cultural and religious diversity seem to have a strong impact on learning groups, sensitive approaches have to be discussed not only on a practical but also on a conceptual level. Continued training for educators and multipliers in the field could help to bridge this gap.

**(d) The idea of experience in culturally and religiously diverse learning spaces**

Regarding results, experiences are crucial to constructive learning environments and children, and educators and parents report experiential learning in culturally and religiously diverse contexts. Interaction with others seems to play a key role when it comes to learning through experience. However, stories, collaborative pedagogic projects, and celebrations

also seem to be very important to the interviewed children. In this context, especially, parents point out that experiences can only be made when the learning environment is regarded as a safe space. Diversity can be experienced when children know that they are fully accepted in their individual backgrounds and worldviews. Again, the question arises of how far experiences in other cultural and religious contexts can be offered without transgressing individual and theological lines (Weber 2014).

Regarding perspectives on socialization, the study indicates that children make experiences in cultural and religious contexts mostly through their family and their peer groups. ECE institutions, however, also seem to play an important role as they can organize moments of experiential learning.

Regarding discussion of an example, some of the interviewed children report visits to religious places, such as temples or mosques. The visits are part of a collaboration between ECE institutions and religious communities. Experience plays an important role during these visits as children vividly report special memories: lights, sounds, smells, tastes, neighborhoods, and people. Collaborations with cultural and religious communities can offer constructive learning potential through experience. However, diversity-sensitive approaches must offer learning experiences for all children, and hence must reflect on possible problems beforehand. Some of the interviewed parents are skeptical about these visits, as they don't want their children to be exposed to the practices of other beliefs. The sample clearly indicates that children, educators, and parents are in favor of experiential learning arrangements. These arrangements, however, have to be developed in close collaboration with children and parents to guarantee that the level of involvement can be chosen by the children.

Regarding outlook, authentic experiences in learning can often connect to the individual contexts of the learner, and can thus lead to a high level of motivation and involvement. It shows that children, educators, and parents appreciate a high level of authenticity that can be created by collaboration with communities. Experiential learning in ECE, however, has to be planned sensitively and accurately to make sure that children are not exposed to negative experiences, especially as the experience of different cultural and religious views, places, stories, or festivals can lead to strong experiences that must be accompanied with pedagogical sensitivity. Against this background, the preparation of experiential learning in diverse contexts has to be very accurate. The reflection of these experiences needs a constructive and sensitive setting, and it should be cared for by educators and parents.

**(e) Actions against discrimination due to cultural and religious belonging**

Regarding the results, this study and the subsequent study from Lichy (2023) both indicate that culturally and religiously diverse learning groups in ECE institutions show problems of discrimination but at the same time can be constructive environments for the development of sensitive attitudes towards diversity. Some educators report discriminatory behavior between children due to political and religious differences. Social status and origin are also mentioned as factors for discrimination. However, most educators describe their learning environments as very tolerant and welcoming. Cultural and religious diversity is taken for granted, can be experienced in everyday interactions, and offers various learning options. In the interviews, the question arises of how discriminatory actions can be prevented. One possible answer, which is discussed by the educators, is the inclusion of parents and families in the educational processes, as many educators report the important role of families in these contexts.

Regarding perspectives on socialization, both studies indicate that families are crucial for the prevention of discriminatory actions. Children should reflect on discriminatory actions together with their whole family. Families should openly discuss possible prejudices in collaboration with ECE institutions.

Regarding the discussion of an example, some educators reported prejudices against refugees. In these cases, children had heard false stories about refugees kidnapping babies. Due to these stories, some children developed anxieties about people with different skin colors. The interviewed educators name skin color, social status, and gender as the main

factors of discrimination. The importance of a close collaboration between ECE institutions and parents is underlined in the interviews multiple times. Prejudices in the family can lead to discriminatory actions in ECE learning environments, as children bring these prejudices to their learning environments as part of their lifeworlds. Therefore, ECE institutions have to reach out to the children's families to reflect on prejudices and overcome reticence.

Regarding outlook, ECE institutions have the responsibility to prevent discrimination and actively act against discriminatory actions. Furthermore, children should have the possibility to identify discrimination and be empowered to act against it. Therefore, the cultural and religious backgrounds of children and their families have to be constructively included in the learning environment. Close collaboration between ECE institutions and families can be a way to overcome prejudices and act against discrimination. The development of an anti-bias attitude must be the main goal of this collaboration.

**(f) Educators and their individual attitudes to cultural and religious diversity**

The results of the study show that educators are crucial for the development of a culturally and religiously sensitive learning environment. They play a key role in the socialization of children and can serve as role models for children and families. Against this background, it is essential to offer educators open spaces and training opportunities to reflect on their individual cultural and religious backgrounds and to discuss these in the context of their role as educators. Therefore, it must be in the highest interest of ECE institutions to offer their educators possibilities to learn about culturally and religiously sensitive approaches and to reflect on their individual attitudes.

Regarding perspectives on socialization, the study indicates that ECE institutions play a major role in children's perception of diversity. Educators can be crucial to the socialization of children and even families.

Regarding the discussion of an example, many educators report that they experience cultural and religious diversity as an important and often challenging topic. The diverse backgrounds of children and families can become confusing and sometimes individual prejudices become a problem. One educator reports about a situation where she offered a parent from Turkey tea and welcomed her in Turkish. However, the parent was Kurd and felt insulted by the well-intended greeting (Kölsch-Bunzen et al. 2016). This shows that the reflection of individual attitudes and prejudices plays an important role in the education of educators. Many educators express their interest in further training, such as anti-bias training.

Regarding outlook, culturally and religiously sensitive education needs experts who can develop diversity-sensitive learning environments and collaboration with families and communities. Therefore, educators need to reflect on their own cultural and religious views, experiences, and contexts.

## 4. Discussion and Outlook

Around the world, societies are changing rapidly and becoming increasingly diversified. Values of inclusion and equity as well as respect for diversity apply to young children and must become essential to ECE institutions. The child's world is shaped through diversity, and children bring their experiences with and attitudes towards diversity to ECE institutions. Thus, it is crucial to deal with diversity in productive and constructive ways, and to establish learning environments that are diversity-sensitive (Gierden-Jülich 2008). The discussion of this research indicates that children can develop cultural and religious sensitivity in ECE institutions when the following prevail:

- They feel accepted and welcomed with their individual cultural and religious backgrounds
- They are enabled to bring their cultural and religious views and experiences to the ECE institution.
- They are encouraged to interact with other children, educators, and communities in culturally and religiously diverse contexts.

- They realize that biases and discriminatory actions are no option in their learning environments.
- They are enabled to experience diverse cultural and religious traditions, festivals, and spaces, and they can choose individual levels of participation and commitment
- Parents and communities are involved as partners of ECE institutions and bring authentic insights to the learning groups.
- Educators show reflective and qualified attitudes toward diversity and are acquainted with diversity-sensitive approaches
- Cultural and religious diversity is part of the institution's concept and can therefore be experienced in everyday learning and living

These requirements seem to be essential if ECE institutions want to become diversity-sensitive spaces of learning and living. The research results indicate that to meet these requirements certain quality characteristics (Kölsch-Bunzen et al. 2016) are crucial:

(1) Educators should reflect on their individual perceptions of 'normality' and possible prejudices within their lifeworlds. Professional self-reflectiveness is the foundation for a diversity-sensitive education.

(2) Prejudices and all forms of discrimination have to be discussed with children and parents. To establish inclusive learning environments all partners have to work hand in hand and act against discrimination. The examination and discussion of language, interactions, and learning materials are part of this process and offer the valuable potential for learning. Possible negative experiences must be sensitively prevented by involving children, parents, and cooperation partners at an early stage.

(3) Cultural and religious diversity have to be constantly reflected from various differentiated perspectives. Tendencies of culturalization have to be identified, discussed, and disestablished. Diversity has to be connected to the ideas of inclusion and equity.

(4) Children's families have to become partners in the process of establishing culturally and religiously sensitive education. As an essential factor of socialization, the child's family should participate in the development of diversity-sensitive learning environments. Children should experience that their families are part of ECE institutions, especially with their individual cultural and religious backgrounds. This can lead to a stronger identification with the institution and a feeling of representation and belonging.

(5) Cultural and religious views and experiences have to be regarded as individual and therefore unique. ECE institutions must focus on every single child's background. Culture-specific and religion-specific attributions often produce prejudices that might lead to discrimination. The 'family culture' or 'family religion' of the child should be the starting point of the learning process.

(6) Opportunities for learning in ECE institutions have to be accessible for all children and therefore need to be reflected under the perspectives of culturally and religiously sensitive approaches. Experiential learning opportunities, especially, should be offered with different levels of commitment. An inclusive learning approach can only be guaranteed when all children find ways to participate.

(7) ECE institutions should be able to justify their work on the foundation of a pedagogical concept that includes culturally and religiously sensitive approaches.

ECE institutions must be developed into spaces of recognition, representation, and belonging in which there is no discrimination. Children need safe spaces of recognition to constructively deal with cultural and religious diversity. Thus, ECE institutions have the chance to become spaces in which children can experience the recognition of cultural and religious differences as something natural and positive. In an inclusive learning space, supported by professional educators, children can learn about others and themselves. Humboldt's idea of alienation from one's own world as a constructive means of learning can be the foundation for a culturally and religiously sensitive education on the grounds of interaction.

**Funding:** This research was funded by Ministerium für Wissenschaft, Forschung und Kunst Baden-Württemberg/Ministry of Science, Research and the Arts Baden-Württemberg (2001–2004).

**Institutional Review Board Statement:** The study was conducted in accordance with the Declaration of Helsinki, and approved by the Institutional Review Board of Freiburg University of Education (1/2001).

**Informed Consent Statement:** Informed consent was obtained from all subjects involved in the study.

**Data Availability Statement:** The data presented in this study are available on request from the corresponding author. The data are not publicly available due to privacy restrictions.

**Conflicts of Interest:** The author declares no conflict of interest.

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
