# Peer review of "Cultural and Religious Diversity in Early Childhood Education Implications of Socialization and Education for the Geographies of Childhood"

_religions, doi:10.3390/rel14040555_

Round 1

Reviewer 1 Report

 I appreciate the opportunity to read and review this manuscript.  I see its importance for the current German context of ECE and the apparent desire to enhance understanding of diversity in ECE and in children's lives.  As for the basic methods used in this research, a clear strength is the multi-informant use of educators, parents, and children as sources of information and insight.  

However, I see many problems with the paper and am not optimistic about its chances for publication.  First, the overall writing style is extremely vague and repetitive.  I think the Introduction could be reduced by at least half, and the rest of the paper seems repetitive in format.  Really, the word "diversity" is used truly excessively, and without elaboration or exploration.  I find pervasive repetition of the paper's basic ideas and recommendations.

A problem is that with such repetition there also is lack of elaboration or depth; I find the treatment of ideas rather shallow and cursory. 

And the shallowness is evident in a key section of the paper--where we are told about the study's methods:  who was interviewed, and how, and what was asked.  Actually, we are told little about the interviews themselves:  What exactly were the educators and parents asked in their interviews?  What were children asked? What kinds of activities were children asked to do to provide their views?  So from a research perspective, the paper is painfully absent of any such key information.  Further, we know little about who these people are--ages, family situations, backgrounds, experiences with children, their own religious background, their own embodied diversity and experiences with diversity.

And this incomplete presentation of information continues with what appears to be some attempt at a "Results" section, but the problem is that we don't actually get any results in any clear, organized form.  To my reading, the findings are something of a mash, a melting pot that dilutes any specifics from parents or children or educators.  We get bland generalities (and ones that are repeated) in the findings.  And from a methodological vantage, any use of content or thematic analysis of the qualitative data from the interviews requires detailed description of how the interview results--that is, transcripts of what the subjects actually said--were analyzed.  Further, what kinds of interrater reliability was achieved, in the analyses of the transcripts? (Were there even transcripts? Or only notes?  We never find out what exactly emerged from the interviews.)  ANd without these details, we don't know how the study was conducted, what people actually said, and whether the coders/raters agreed on their interpretations of the meaningful content and themes in their subjects' comments.  In short, the methods and results are woefully inadequate.

Then we arrive at the vague interpretations of the findings, where, to my eye, the only valuable information comes in the specific examples that are presented.  These are few and far between, given the three types of informants' interviews.  I also note that there is virtually no use of literature from any field--religious education, anthropology, developmental psychology, etc.--to frame and interpret those examples. 

The conclusions and recommendations at the end of the study are also problematic.  First, it's not clear how they derive from any "findings" because the "findings" are so briefly and incompletely presented.  Second, the final section contains what strikes me as overly prescriptive demands about what schools and parents "should" do.  I am not German, and know nothing, really, about its current political or ideological climate.  But the recommendations strike me as overly dogmatic and rigid, making little room for what would probably be negotiated meanings and potential steps to progress by policy-makers, educators, and families. 

Author Response

Thank you very much for your review.

Please find my answers below (please note: I discuss the remarks I found helpful and constructive):

I think the Introduction could be reduced by at least half, and the rest of the paper seems repetitive in format.

  • I partially share this assessment
  • Please see changes in the revison

What exactly were the educators and parents asked in their interviews?  What were children asked? What kinds of activities were children asked to do to provide their views? 

A discussion of the questionnaires of all actors is – unfortunately ! - beyond the scope of this article.

However, I added some information about this in the revision.

Further, we know little about who these people are--ages, family situations, backgrounds, experiences with children, their own religious background, their own embodied diversity and experiences with diversity.

  • Relevant information: 210-240.

To my reading, the findings are something of a mash, a melting pot that dilutes any specifics from parents or children or educators. 

  • This paper focuses on results from the triangulation of perspectives. It is beyond the scope of this paper to present and discuss specific results from all perspectives.

Were there even transcripts? Or only notes?  We never find out what exactly emerged from the interviews.

  • See 210-220

But the recommendations strike me as overly dogmatic and rigid, making little room for what would probably be negotiated meanings and potential steps to progress by policy-makers, educators, and families.

  • The recommendations are impulses derived from the results of the study. They are within an area that is common and desired in a scientific context.

Again thank you for your thoughts and recommendations.

Reviewer 2 Report

There were 200 children and parents in the study. I would be interested in knowing their religious/cultural backgrounds. Did the families self-select to be in the study? 

Excellent description of methodology.

What questions were asked in the semi-structured qualitative interviews? How did the researchers begin the discussion with children, parents, and educators? 

The manuscript would benefit from more specific examples of religious stories or celebrations (other than Christmas) that children and parents referenced. The reported results were quite general. Did any teachers encounter experiences in which they questioned the appropriateness of a topic (other than prejudice toward refugees) that children mentioned? 

I wonder about the recommendation that young children might choose the degree of religious content that they want to discuss. How would a four- or five-year-old make such a decision? Would families be consulted about the religious content of various activities and be part of the decision-making? 

I am confused by the numbering on lines 398-402.

On lines 425-427 the authors mention collaboration between ECE institutions and families as a way to overcome prejudice and discrimination. What else can happen within the classroom to help children experience alternate views? Children's literature? Storytelling? 

What are expectations regarding religious diversity in Germany? In the U.S. it is clear that teachers can teach about different religions, but should not advocate religion over non-religion or advocate one religion over others. 

Author Response

Thank you very much for your very helpful and detailed review!

There were 200 children and parents in the study. I would be interested in knowing their religious/cultural backgrounds. Did the families self-select to be in the study? 

I discuss this in the review (223)

What questions were asked in the semi-structured qualitative interviews? How did the researchers begin the discussion with children, parents, and educators? 

A discussion of the questionnaires of all actors is – unfortunately ! - beyond the scope of this article.

However, I added some information about this in the revision.

The manuscript would benefit from more specific examples of religious stories or celebrations (other than Christmas) that children and parents referenced. The reported results were quite general. Did any teachers encounter experiences in which they questioned the appropriateness of a topic (other than prejudice toward refugees) that children mentioned? 

Thank you for this remark. I chose the examples as they are connectable and demonstrate overarching interrelationships. I agree that it would also be interesting to discuss more specific examples but I think this is beyond the scope of this article.

I wonder about the recommendation that young children might choose the degree of religious content that they want to discuss. How would a four- or five-year-old make such a decision? Would families be consulted about the religious content of various activities and be part of the decision-making? 

Thank you – I discuss this in the revision. 503ff / 289ff

I am confused by the numbering on lines 398-402.

So am I. I got rid of it – thank you!

On lines 425-427 the authors mention collaboration between ECE institutions and families as a way to overcome prejudice and discrimination. What else can happen within the classroom to help children experience alternate views? Children's literature? Storytelling? 

Probably yes, but this was not part of the results.

What are expectations regarding religious diversity in Germany? In the U.S. it is clear that teachers can teach about different religions, but should not advocate religion over non-religion or advocate one religion over others. 

This is hard to answer and beyond the scope of the article. I think that Germany is on its way to a more diversity sensitive idea of education. Religious institutions play an important role as partners of the state and have to be involved in this process.

Again, thank you very much for your thoughts and recommendations!

Reviewer 3 Report

See attached file.

Author Response

Thank you very much for your very helpful and detailed review!

1) It is not until line 349 that we fully see that the
empirical research is conducted in German ECE
intitutions. Why not mentioned earlier? 

Done.

2) It would be
meaningfull to add a specific cultural example:
what cultural community, what cultural
experience of children?

see: 302-309

3) Your point is clear. But can’t you use a more
recent reference to underline this point?

Frenkel-Brunswick established the idea. Gauss (2021) discusses it in the context of today's society. 

4) Please reflect on the following question: is the
role of agent of the child, and especially the
young child, not an overestimation of its
competence?

Important point - thank you. 

Done.

5) Line 73-97 From a theoretical point of view you underline a
crucial point. However, can you add some more
recent references tot his point?

Done (Grümme, 2017)

6) Line 101-106 I don’t see why a reference to Humboldt (1793!)
is worthwhile. Please use more recent
theoretical views for this same, important point.

One of the founders of the idea of constructivism (in the context of pedagogy) and crucial to the discussion in Germany up until today. The following reference (2021) puts Humboldt's ideas in context. 

7) Line 139-146 * There seems to be an overlap between
dimension 4 and 5. When dimension 4 includes
‘existential questions’ and traditions this can
also include the question for a higher being.
Why is this 5th dimension not included in the
4th?

- The Dimensions are disucced according to the Boschki (2003) - the selectivity of Boschki's model is not very high as all dimensions are interconnected. It still works as a helpful framework.

Why do you start the decription of the 5th
dimension whith a reference tot he Christian
anthropology and Rahner? The following is
much more crucial

  • I worked on this

8) Line 113-152 You might consider building your strong focus
on child-centreness and relation upon
theoretical sources that underline a
hermeneutical-communicative point of view.

I added further references and worked on the dimensions. 

9) Line 185
What do you mean with ‘partners in the community’, and why are they not included in the empirical research?

I discuss/explain it in the revision.  An inclusion in the research was not possible at the time due to a lack of funding. 

10) Line 214
Why are capitals used for Children, Educators, Parents?

Changed

11) Line 362
What is meant by ‘projects’?

Explained

12) Line 366-368
Who askes this question? Parents, educators, children? Or the author? Very essential question in the context of this article!

It is discussed by Weber 2019. The reference is in the revision now.

What is, in this context, the meaning of ‘negative experiences’ (line 389-390)?

I discuss this in 393

Why no elaboration on the question when ‘individual and theological lines are trangressed’ and an experience is ‘negative’? And why is this crucial question and the attention for ‘negative experiences’ not described in the concluding remarks (line 487-515)? 

I discuss this in the concluding remarks now. Thank you for pointing this out.

 Line 405-407

Why asking a question and providing an answer in a section that discusses results? Also in other results paragraphs.

I tried to make this clear in the revision: Within the interviews questions came up and were discussed by the participants. Thus their answers are part of the results.

Again, thank you very much for your thoughts and recommendations!